# Developing a behavioural intervention package to identify and amend incorrect penicillin allergy records in UK general practice and subsequently change antibiotic use

Marta Santillo ,[1] Marta Wanat,[1] Mina Davoudianfar,[1] Emily Bongard ,[1] Sinisa Savic ,[2] Louise Savic,[2] Catherine Porter,[3] Joanne Fielding,[3] Christopher C Butler,[1] Sue Pavitt,[4] Jonathan Sandoe,[3] Sarah Tonkin-Crine ,[1] The AlABAMA team

[1]Nuffield Department of Primary Care Health Sciences, Oxford University, Oxford, UK
[2]Faculty of Medicine and Health, University of Leeds, Leeds, UK
[3]Healthcare Associated Infection Group, University of Leeds, Leeds, UK
[4]University of Leeds, Leeds, UK

**Correspondence to**
Dr Marta Santillo;
marta.santillo@phc.ox.ac.uk

## ABSTRACT

**Objectives** To develop a behavioural intervention package to support clinicians and patients to amend incorrect penicillin allergy records in general practice. The intervention aimed to: (1) support clinicians to refer patients for penicillin allergy testing (PAT), (2) support patients to attend for PAT and (3) support clinicians and patients to prescribe or consume penicillin, when indicated, following a negative PAT result.

**Methods** Theory-based, evidence-based and person-based approaches were used in the intervention development. We used evidence from a rapid review, two qualitative studies, and expert consultations with the clinical research team to identify the intervention 'guiding principles' and develop an intervention plan. Barriers and facilitators to the target behaviours were mapped to behaviour change theory in order to describe the proposed mechanisms of change. In the final stage, think-aloud interviews were conducted to optimise intervention materials.

**Results** The collated evidence showed that the key barriers to referral of patients by clinicians were limited experience of referral and limited knowledge of referral criteria and PAT. Barriers for patients attending PAT were lack of knowledge of the benefits of testing and lack of motivation to get tested. The key barriers to the prescription and consumption of first-line penicillin following a negative test result were patient and clinician beliefs about the accuracy of PAT and whether taking penicillin was safe. Intervention materials were designed and developed to address these barriers.

**Conclusions** We present a novel behavioural intervention package designed to address the multiple barriers to uptake of PAT in general practice by clinicians and patients. The intervention development details how behaviour change techniques have been incorporated to hypothesise how the intervention is likely to work to help amend incorrect penicillin allergy records. The intervention will go on to be tested in a feasibility trial and randomised controlled trial in England.

### Strengths and limitations of this study

► Intervention development was conducted using a systematic and iterative process
► Evidence-based, theory-based and person-based approaches were used which consisted of three iterative stages: collating the evidence on views and experiences of penicillin allergy testing, intervention planning informed by theory and optimisation of intervention materials
► These approaches were used to increase the feasibility and acceptability of the intervention to its target users (general practitioners and patients)
► This approach clearly maps how the intervention components are hypothesised to change clinician and patient target behaviours
► The views of other general practice prescribers (such as nurse prescribers and pharmacists) should be considered in future research

## BACKGROUND

Incorrect penicillin allergy records are common, as side effects and symptoms of illness can be mistaken for allergic reaction symptoms.[1] About 6% of the UK general practice population has a record of a penicillin allergy but fewer than 10% of these patients are likely to be truly allergic.[2] Penicillin allergy refers to an allergy to a whole group of antibiotics (penicillins), which are first-line antibiotics for many common infections. As a result of incorrect penicillin allergy labels, a significant portion of the population is denied effective antibiotics. Moreover, penicillin allergy records are also linked to antimicrobial resistance: evidence suggests that patients with a penicillin allergy label are more likely to be prescribed broad



spectrum antibiotics and to acquire antibiotic resistance infections.[3 4]

Penicillin allergy testing (PAT) is already provided in the National Health Service (NHS) in specialist clinics[5] and offers the opportunity to confirm or discount a penicillin allergy label. This service is only available to a subset of patients following National Institute for Health and Care Research (NICE) guidance.[6] Current assessment is performed over two clinic visits.

The Allergy Antibiotics and Microbial resistance (AlABAMA) Study aims to develop a behavioural intervention package for UK general practice to effectively amend incorrect penicillin allergy records. The intervention package is designed to target both general practice clinicians and patients with a suspected incorrect penicillin allergy record. It introduces a pre-emptive 'penicillin allergy assessment pathway' (PAAP) that targets patients assessed as 'low risk' of true allergy, who are not at risk of anaphylaxis or other severe adverse reactions, and it aims to streamline the test process by undertaking patient history screening in general practice (stage 1) and introducing an efficient one-stop procedure at a hospital clinic for the PAT. The test includes either a skin test (ST), testing penicillin solution on the forearm (stage 2) and oral challenge test (OCT), taking doses of penicillin solution over time (stage 3), or just OCT depending on the individual patient. Following the PAT, patients and practices would receive confirmation of a patient's allergy status. The AlABAMA intervention will be trialled in a national randomised controlled trial in winter 2020–2021. It will test how corrections of incorrect penicillin allergy records and subsequent penicillin prescribing influence recovery and patients taking antibiotics for infections.

This paper describes the planning, development and optimisation of the AlABAMA PAAP intervention package.

## METHODS AND RESULTS
### Intervention planning methodology
We followed an integrated approach to intervention development that combines theory-based, evidence-based and person-based approaches.[7–9] This approach has been successfully used in the development of a variety of behavioural interventions including for reduction of antibiotic prescriptions in European general practice.[10]

### Patient and public involvement (PPI)
The AlABAMA programme grant involved PPI members, as co-applicants, and study advisors from the start so research questions were informed by their input. The results of all work from the AlABAMA programme will be disseminated with the help from our PPI co-applicants and our existing PPI networks towards the end of the programme grant.

The AlABAMA intervention package was developed in three stages described below. The methods and key results for each stage are presented below.

### Stage 1: collating and analysing evidence
#### Rapid review methods
The full rapid review is available elsewhere.[11] Rapid review is a form of synthesis that supports the review of existing evidence in a timely manner.[12] It aimed to explore clinicians' and patients' views and experiences of PAT services.

#### Results
The review identified only two studies that reported patients' views of PAT. The first study sought patients' views on undertaking a three-step testing procedure consisting of prick testing, intradermal testing and oral challenge.[13] It found that patients thought that PAT provided a valuable medical information; however, this was measured using a single-item questionnaire ('Do you think undergoing testing for penicillin allergy provides important information for your penicillin history?'). The second study recruited patients who attended general allergy clinic and were given a brief five-question survey examining their views of PAT.[14] This study found that patients were not aware of PAT and had limited understanding of penicillin allergy, but also expressed their interest in being tested.

The review also identified six studies on clinicians' views of PAT[13 15–19] three of which specifically focused on barriers and facilitators to using PAT.[13 17 19] One of the studies included both patient and clinician views.[19] Clinicians reported several barriers to referring patients for PAT. These included difficulties establishing the allergy history, lack of knowledge on referral processes and organisational pressures making allergy testing a low priority.[13 17 19] A number of clinicians and patients reported being reluctant to prescribe or consume penicillin after a negative PAT result.

#### Qualitative methods
Full details of the qualitative work are available elsewhere.[20] Two qualitative studies were undertaken by MW; one interviewing 19 general practitioners (GPs) and the second interviewing 31 patients with a penicillin allergy record (16 with previous experience of PAT). The aim was to identify clinicians' and patients' views on the barriers and facilitators for PAT and antibiotic use after a negative test. Semistructured interviews were conducted over the phone by an experienced qualitative researcher (PhD qualified) with substantial previous experience of conducting qualitative research. Clinicians were identified in general practices and by the local microbiology services. Patients were identified from a general adult hospital allergy clinic and from general practices in the same geographical area. Interview topic guides were informed by the existing literature, which indicated potential barriers and facilitators to target behaviours.[20] Written informed consent was obtained from all participants. An inductive thematic analysis approach was used to analyse data.[21]

## Results

Clinicians reported that they often felt that penicillin allergy records were incorrect, however, reported uncertainty about how to identify patients with true penicillin allergy and were reluctant to amend medical records without objective evidence. Penicillin allergy status was not seen to be a major problem in general practice due to the availability of alternative antibiotics and clinicians struggled to identify the risks of incorrect allergy records. Clinicians were seen to differ in their consultation styles when speaking to patients about their antibiotic prescribing decisions and allergy status. They reported lack of experience of PAT services and the need for more information on referral criteria. Regarding the process of changing a patients' record after a negative test result, clinicians reported being happy to update medical records on directions from secondary care but were unsure who was responsible for making sure that patients understood allergy test results.

The majority of patients who were motivated to get tested had experienced a negative consequence of having a penicillin allergy label (such as limited availability of antibiotics they could use). Patients reported concerns about the possibility of having an allergic reaction during PAT; the degree of the severity of their previous reactions affected how apprehensive they were about the test. Moreover, when the test was perceived as more invasive, for example, the OCT compared with the ST, patients reported being more concerned about PAT. Patients were also concerned about how much they would be monitored during the test and highlighted the importance of being informed, kept safe and observed by qualified professionals. Some patients reported being unsure about a negative test result and feeling anxious about taking penicillin if prescribed, as they doubted whether the test result was accurate.

### Expert discussions

In addition to our core team (which included three psychologists), the wider clinical research team included a consultant immunologist, a consultant microbiologist, a consultant anaesthetist, a GP and professors with expertise in applied health research. As part of this first stage, we consulted our wider clinical research team using monthly teleconferences and emails to gain their feedback on several aspects of the intervention development.

## Results

The expert discussions informed the interpretation of the evidence collated in the rapid review and the qualitative studies, the development of early iterations of the intervention materials and the development of the initial intervention plan and components.

### Stage 2: intervention planning and development
#### Creating guiding principles

In line with the person-based approach,[7] brief guiding principles were created to be consulted through the whole intervention development process. This ensured that the intervention met the original objectives. As a first step to creating guiding principles, characteristics and behavioural needs of the target users were identified based on the findings of the rapid review, qualitative interviews and expert discussions. The second step involved identification of intervention objectives and key design features, which would address these needs.

## Results

Table 1 presents the AlABAMA guiding principles. These focused on increasing confidence to refer and attend for PAT and increasing motivation to prescribe/consume penicillin following a negative PAT result. Guiding principles also included increasing clinician confidence in discussing penicillin allergy with patients and improving communication between primary and secondary care about penicillin allergy status. Lastly, the intervention needed to present the PAAP as reliable and trusted and provide accessible and easy to use materials for clinicians and patients.

### Behavioural analysis

The aim of the behavioural analysis was to use behaviour change theory to describe the content of the AlABAMA intervention package and map the evidence from the rapid review, qualitative studies and expert consultations. We used a model of behaviour known as capability, opportunity, motivation, behaviour (COM-B).[22] According to this model, behaviour is a result of these three constructs. Changing behaviour requires overcoming any presenting barriers in each of these areas. The behaviour change wheel (BCW)[23] offers a systematic way to develop interventions to change each of these constructs. A more specific description of elements which can be delivered within intervention components is provided in a taxonomy of 93 behaviour change techniques (behaviour change technique taxonomy, v1; BCTTv1).[24]

## Methods

The first step of the behavioural analysis process was to define the problem in behavioural terms, which resulted in the identification of the four specific target behaviours.

As a second step, we identified what needed to change using the Theoretical Domain Framework (TDF),[25] which includes 14 domains such as knowledge, skills and beliefs about capabilities. Each domain of the TDF relates to a COM-B component. This step allowed us to identify target behaviours, their barriers and facilitators, and how intervention components would support desired behaviour change based on evidence collated in stage 1.

Finally, the intervention components were mapped to the TDF framework and the BCW referring to the BCTv1.[24] This produced a list of TDF barriers to the target behaviours, target constructs (what needs to change for the behaviour to occur), intervention functions (ways an intervention can change behaviour)

**Table 1**  Guiding principles for the AlABAMA intervention package

| Intervention design objectives | Key features |
|---|---|
| To present the PAAP as a reliable and trusted approach to confirm allergy status | ► Present the PAAP as a trusted/scientific/official way to get a confirmed PAT result, for both clinicians and patients |
| To increase clinician's confidence in referring patients, and patients' motivation to attend, for PAT | ► Provide evidence on the potential harms of an incorrect penicillin allergy records and the process of testing in the information pack<br>► Provide opportunity to address patients' concerns about potential benefits and risks of testing by both clinicians and allergists during consultations before the allergy test and during the appointment at the immunology clinic |
| To motivate clinicians/patients to prescribe/take penicillin following a negative PAT test result | ► Provide information to clinicians about the clinical meaning of a test result and its implication as part of the information pack<br>► Provide information to patients about the PAT (process) and the potential benefits of being able to take penicillin in the pretest intervention booklet |
| To increase clinicians' ability to discuss PAT with patients | ► Provide training for general practice clinicians about penicillin allergy and PAT, including its accuracy, implications, benefits |
| To improve communication between primary and secondary care so that allergy status records are correct | ► Provide a clear and consistent approach to delabelling with support from colleagues in secondary care and preventing relabelling in the future<br>► Provide information about who is responsible for ensuring patients understand the results and for updating the medical records during the site training and in the information pack<br>► Provide clear and precise documentation of side effects during future courses of penicillin in the information pack |
| To provide easily accessible tools that are suitable for use by general practice clinicians | ► Make interventions materials for clinicians accessible, short, easy to follow, easy to implement and not increase workload |
| To provide easily accessible resources for patients | ► Make patients materials brief, easy to read, accessible<br>► Make materials easy to carry with them at all time and that provide evidence of the results of their test |

PAAP, penicillin allergy assessment pathway; PAT, penicillin allergy testing.

and behaviour change techniques used for each of the barriers/facilitators.

*Results*

The full behavioural analysis is presented in online supplemental materials S1. First, we identified barriers and facilitators to referral of low-risk patients to PAT and patient attendance to PAT. The analysis highlighted that both clinicians' and patients' knowledge and perceptions of penicillin allergy and test procedures could be modified; information needed to be supported by scientific evidence for clinicians and patients to be reassured that the test was safe. We designed a resource for clinicians entitled 'Penicillin Allergy Testing: Information for general practice', which contained information on penicillin allergy and PAAP procedures. As part of the AlABAMA Trial, this will be supported by site training and working instructions, which provide practical guidance on screening patients and referral to PAT (relevant BCTs for clinicians were 'information about antecedents' and 'information about health consequences').

For patients we developed two patient booklets, one to be provided prior to PAT and one following a negative test result. All patients, on entering the trial, will have a consultation with a GP to answer questions and address concerns about PAT. We developed a patient booklet entitled 'Penicillin Allergy Testing: going for a test', which included information on PAAP procedures and PAT safety (relevant BCTs for patients were 'information

about health consequences' and 'feedback on outcomes of the behaviour').

The barriers to the prescription and consumption of first-line penicillin following a negative test result were patient and clinician beliefs about the accuracy of PAT and whether taking penicillin was safe. Clinicians also needed reassurance that colleagues saw delabelling as beneficial and resources to support them in changing incorrect penicillin allergy records. We developed a second patient booklet entitled 'Penicillin Allergy Testing: a negative test result', which contained information about which antibiotics patients could safely take in the future following a negative test result, a negative result intervention card and a result letter which confirmed the patient allergy status to penicillin (relevant BCTs were 'social support' and 'restructuring of the social and physical environment'). As part of the trial, clinicians will receive working instructions, which contain guidance on how to change the patient allergy label in medical records, result letter which confirms the patient allergy status to penicillin and an electronic pop-up, which includes a reminder of the patient's new allergy status (relevant BCTs were 'feedback on outcomes of behaviour' and 'adding objects to the environment').

*Logic modelling*

The next step included the development of a logic model, which summarised the behavioural analysis, providing a diagrammatic representation of the hypothesised

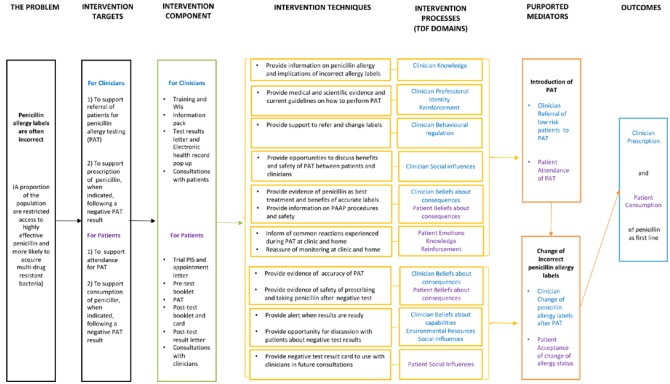

**Figure 1** AlABAMA logic model. PAAP, penicillin allergy assessment pathway; PIS, participant inforamation sheet; TDF, Theoretical Domain Framework; WI, Working Instruction.

processes and causal pathways from the intervention components to the desired outcomes.[26][27]

### Methods
The research team opted for a process oriented iterative logic model, which was refined during the whole intervention development stage.

### Results
The logic model (figure 1) included four components:

#### Intervention components and techniques
Intervention components were organised based on the two target groups (clinicians and patients). Intervention techniques summarised the BCTs used as part of the intervention, which were identified in the behavioural analysis.

#### Intervention processes
These were the psychological factors that explained the relationships between the intervention components and the outcome of the intervention. These processes were described in terms of TDF domains. Each intervention technique was hypothesised to mainly affect one of these processes. As part of the intervention, clinicians would receive information on penicillin allergy and implications of incorrect allergy records in order to increase their knowledge of PAT and allergies. In addition to this, providing medical and scientific evidence and current guidelines on how to perform PAT would change their professional identity related to their role in the referral process and act as reinforcement to increase their motivation to refer patients in the future. Regarding the processes affecting patient's attendance of PAT, providing evidence of penicillin as best treatment and the safety of PAT would increase their beliefs of positive consequences of accurate records; informing them of common reactions during PAT and reassuring them of monitoring at the clinic and at home would increase their knowledge on PAT, decrease their negative emotions, such as anxiety, and act as positive reinforcement to attend the clinic.

Both patients and clinicians would receive evidence of the accuracy of PAT, and of safety of prescribing penicillin after a negative test result, in order to increase their belief about positive consequences of prescribing penicillin after a negative test result. Finally, providing a negative test result card to use with clinicians would affect patients' expectations of social influences regarding clinician's acceptance of the test result.

#### Purported mediators
These are the target behaviours of the intervention, which directly affect the outcomes. In the logic model, the assessment of potential incorrect penicillin records was operationalised as the referral of low-risk patients to PAT and patient attendance at PAT. The introduction of the PAAP was hypothesised to affect the change of incorrect penicillin allergy records (clinician changing medical records and patient acceptance of change of penicillin allergy status), which would ultimately affect the consumption of penicillin.

#### Outcomes
The behavioural outcomes of the model were the prescription and consumption of first-line penicillin when indicated.

### Stage 3: optimising the intervention materials
The development of the intervention materials was part of an iterative process that was initially informed by the findings of the rapid review, qualitative interviews and expert consultations. Once the first drafts of the intervention materials were developed, a think-aloud approach was used with clinicians and members of the PPI group for the aim of feedback on intervention materials. As this did not constitute a formal qualitative evaluation, ethical approval was not required. This series of think-aloud consultations with clinicians and PPI representatives was conducted in order to gain their insights on the content, usability and format of the clinician 'Penicillin Allergy Testing: Information for general practice' leaflet and two patient booklets ('Penicillin Allergy Testing: going for a test' and 'Penicillin Allergy Testing: a negative test result'). Subsequently to the feedback received by the participants in the think-aloud consultations and discussions in further expert consultations, the intervention materials were optimised. Verbal consent was obtained from all participants, who agreed to take part in the consultations and for notes to be taken. Consultations were not recorded and interviews were only used to refine intervention materials.

### Think-aloud consultations with GPs
#### Methods
Think-aloud telephone consultations were conducted with six participants by MW; two additional participants provided feedback via email. Four of the participants had previously taken part in the qualitative study on GP views on the barriers and facilitators for PAT and antibiotic use after a negative test (mentioned in the section

Qualitative methods); four additional participants were recruited using convenience sampling. Consultations were not recorded, but the interviewer took detailed notes. Consultations focused on GP views of each section of the 'Penicillin Allergy Testing: Information for general practice' (online supplemental materials S2).

## Results

The leaflet was well received. Participants reported that it was informative, useful and generally easy to read. The participants perceived it not only as information for themselves, but also a tool to use in a consultation with patients. Some participants felt that they knew about the consequences of incorrect penicillin allergy record; and therefore, the leaflet could be shortened. Most participants understood the testing stages; however, a couple of participants were confused about which stages of the test patients could skip. One participant wanted exact doses of penicillin specific (rather than just amounts). Regarding the section on patient discussions, some clinicians felt that there was no need to discuss the test with patients. Participants queried whether being tested with amoxicillin meant that the patient could now take all penicillin-based antibiotics and wanted more information.

Clinicians' feedback was collated and organised in a 'table of changes' (online supplemental materials S3) where suggested changes were listed and given a level of priority for that change, following the Must Should Could Would (MoSCoW) framework,[28] and the source of the suggested change (expert opinion, research team, clinical research team, literature review). Changes to the 'Penicillin Allergy Testing: Information for general practice' leaflet included changes to the title, to the exact doses of penicillin given to the patients during the test, information about side effects and information about which antibiotics patients with a negative test result can take safely.

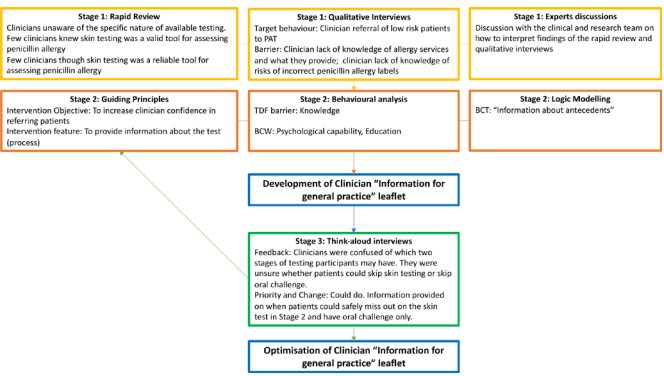

**Figure 2** Example of intervention development for clinician materials. BCT, behaviour change technique; BCW, behaviour change wheel; PAT, penicillin allergy testing; TDF, Theoretical Domain Framework.

## Think-aloud consultations with patients
### Methods

Think-aloud consultations were conducted with seven patients (three with experience of PAT and four with no experience) by MW. Five of these patients had previously taken part in the qualitative study on views on the barriers and facilitators for PAT and antibiotic use after a negative test (mentioned in the section Qualitative methods). These participants were asked at the end of the initial interview whether they would like to comment on the intervention materials at a later date; the additional two participants were identified using convenience sampling. Again, consultations were not recorded, but the interviewer took detailed notes. Consultations asked their views about the two patient booklets ('Penicillin Allergy Testing: going for a test' and 'Penicillin Allergy Testing: a negative test result') and the intervention card (online supplemental materials S2).

### Results

The booklets and intervention card were very well received by the participants. Participants considered the booklets to have the right amount of information and felt they were generally easy to read. Patients reported that the booklets convinced them that going for a PAT could be beneficial. They felt that they could relate to the description of how people were given penicillin allergy labels. Patients thought the description of the test was clear and they knew what to expect. Statistics about the prevalence of allergy were not always understood by the participants, as the participants often thought that 1 in 10 people are allergic and they wanted a more visual presentation of this key information. Participants were unsure what narrow and broad spectrum antibiotics were and did not recognise Methicillin-resistant Staphylococcus aureus (MRSA) abbreviation. Participants did not always know that penicillin is more than one antibiotic. The participants wanted to have a separate paragraph on what could happen during the test and what could happen during 3 days of taking penicillin at home. They also wanted reassurance that 3 days would be enough to detect delayed reactions. The participants wanted more reassurance that after being tested with one type of penicillin (eg, amoxicillin), it would mean that they could safely take all penicillin antibiotics. The participants were slightly concerned about the risk of allergic reaction in the future (despite negative test results).

Patient feedback was collated in a table of changes. Changes made to the booklets were the selection of new images of patients for the front cover, inclusion of definitions of narrow and broad spectrum antibiotics, and reassurance that 3 days of oral challenge would be enough to detect delayed reactions to penicillin.

### Intervention components

The development and optimisation of clinician and patient intervention materials was part of an iterative process. Figures 2 and 3 present the example of this

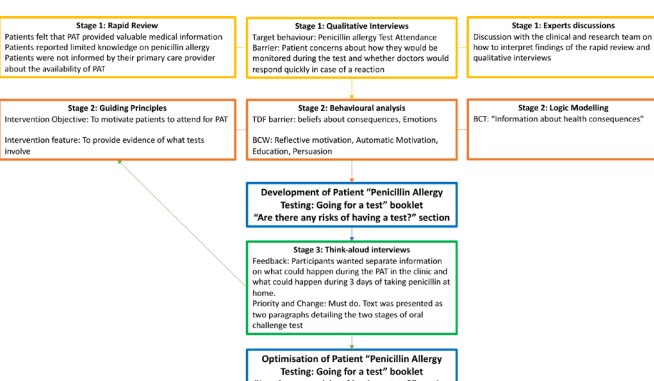

**Figure 3** Example of intervention development for patient materials. BCT, behaviour change technique; BCW, behaviour change wheel; PAT, penicillin allergy testing; TDF, Theoretical Domain Framework.

process for the development of one section of the 'Penicillin Allergy Testing: going for a test' patient booklet and the 'Information for General Practice' leaflet for clinicians.

All Working Instructions (WI) developed to support clinicians and research nurses activities as part of the AlABAMA intervention package were shown to a group of clinicians to gain their feedback on content and layout. Among the clinicians who provided feedback there were two practice managers, and one nurse. Their overall feedback was positive and the main changes to the intervention materials included the identification of the best way of updating the patient's medical records after PAT, and the introduction of screenshots of the medical record in the working instructions.

All participants' letters (patient appointment letter, patient result letter, clinician result letter) were developed among the wider clinical and research team in order to make them effective in motivating patient to attend the PAT and in order to persuade clinician to change patient records and prescribe penicillin after a negative test result, and patient to consume penicillin after a negative test result.

At the end of the intervention development stage, a description of the intervention was completed following the Templete for intervention description and replication (TIDieR)[29] guidance (online supplemental materials S4) together with a description of the intervention components for clinicians and patients (table 2). We have followed the Guidance for reporting intervention development studies in health research (GUIDED)[30] checklist (online supplemental materials S5), when reporting intervention development in this paper.

## DISCUSSION

This paper describes the development of the AlABAMA intervention package, which aimed to change behaviours that facilitate PAT and subsequently amend incorrect allergy records in English general practice. Such changes in clinician and patient behaviours have potential

to significantly impact on antibiotic prescribing and consumption. The approach used here has previously been used for the development of behaviour change interventions targeted to reduce antibiotic prescribing by clinicians in primary[31] and secondary care,[32] but it is the first time that this approach has been used to develop an intervention to amend incorrect penicillin allergy records. The transparency of the intervention development process will inform intervention developers on how this methodology could be used in different contexts, and will facilitate the comparison with other interventions which have used similar processes.

The evidence collated from the rapid review and qualitative interviews allowed in-depth understanding of participant needs (lack of knowledge of PAT, lack of knowledge of negative consequences of allergy labels) and which behavioural influences needed to be modified as part of the intervention. The mapping of these behavioural influences and the AlABAMA intervention package components to behaviour change theory and the logic model allowed a transparent reporting of the psychological processes that are hypothesised to explain the effect of the intervention components on the outcomes. More specifically it highlighted the BCTs used in each component of the AlABAMA intervention package and how they addressed the barriers identified in the rapid review and qualitative work. It also explained which psychological mechanisms were hypothesised to change by the intervention components (clinicians' knowledge on penicillin allergy and PAT procedures, patients' beliefs about positive consequences of taking the PAT) and which target behaviours (referral and attendance to PAT, prescription and consumption of penicillin first line) would likely affect the intervention outcomes. Think-aloud interviews with patients and clinicians, and in-depth feedback from the clinical research team on the intervention materials, highlighted changes that needed to be made in order to increase their acceptability and potential effectiveness.

The AlABAMA intervention package targeted clinician referral of patients for PAT and updating incorrect penicillin allergy records; factors previously identified in previous qualitative research as barriers to effective penicillin allergy delabelling.[11–20] Recent exploration of clinician reported barriers and enablers to identifying and delabelling hospital inpatients with incorrect penicillin allergy records has highlighted the need to introduce patient education concerning the risks of avoiding penicillin.[24] Inconsistencies in the management of penicillin allergic patients were reported, together with a lack of time to discuss allergy testing, and the need to improve communication between primary and secondary care about patient allergy status, as well as updating of patient medical records.[33] A previous exploration of views about implementing delabelling of patients ahead of elective surgery identified barriers to implementing it on a large scale, such as human factors linked to anxiety and financial implications. These human factors were: lack of interest from patients in undertaking an allergy test; lack

**Table 2** Overview of the key intervention components of the AlABAMA intervention package

| Intervention components | Description |
| --- | --- |
| **For clinicians** | |
| "Penicillin Allergy Testing: Information for general practice" leaflet | Information leaflet that includes evidence-based information to increase knowledge about PAT and motivation to refer patients to PAT and prescribe penicillin after a negative PAT result |
| Electronic health record pop-up | Electronic pop-up on medical records. which tells clinicians that patient has had a negative test result and that their allergy status needs to be amended |
| Allergy result letter | A letter is sent to the GP to inform them of the patient's penicillin allergy test result, including details on which test(s) they undertook, whether it is safe or not to prescribe penicillin in the future and instructions on how to change the allergy label |
| Consultation with patients | Discussions with patients to check eligibility for the trial and to answer any queries about the trial and PAAP testing |
| Site training and working instructions | Training in trial procedures delivered to GP leads including provision of information on penicillin allergy delabelling, the referral process, the three stages of the PAAP and the interpretation of test results |
| **For patients in intervention arm** | |
| 'Penicillin Allergy Testing: going for a test' booklet | A booklet to inform patients, in the PAAP intervention arm, about incorrect allergy records, how they may benefit from having a penicillin allergy test and what the test involves |
| Penicillin allergy test appointment letter | The letter includes information on the PAAP procedures in hospital, including pretest assessment and monitoring during the test and at home |
| Penicillin allergy assessment pathway (PAAP) | Appointment at the immunology clinic for patients in the PAAP intervention arm. At the appointment, patients will complete stage 1, stage 2 and/or stage 3 of the PAAP pathway |
| Allergy test result letter | Patient will receive a letter with information on the result of their allergy test and whether it is safe or not to take penicillin in the future |
| 'Penicillin Allergy Testing: a negative test result' booklet | A booklet on the reliability of the test results and consequences of a negative test result (sent with allergy test result letter). |
| Post-test intervention card | Laminated credit card-sized card that says which test the patient has had and confirms the negative allergy result |
| **For all patients in control and intervention arm** | |
| An invitation letter and participant information sheet | An invitation letter and participant information sheet on the purpose of the trial and what the research study would involve for patient participants |
| Discussion with clinicians | Discussion with clinicians about attending the test to ask any queries around the benefits of taking the test and why removing the incorrect record might be good |

GP, general practitioner; PAT, penicillin allergy testing.

of acceptance of the test result among clinicians; high proportion of patient relabelled themselves after a negative testing for penicillin allergy or relabelling by healthcare professionals. The financial barrier was significant despite long-term cost benefit, as there is an upfront cost to perform the test.[34]

A limitation of the AlABAMA intervention package is to how widely applicable it might be. Allergy services across the UK vary significantly, and access to specialist testing ranges widely. The AlABAMA programme will focus on one geographical area (the North of England), which will be covered by the specialist allergy unit participating in the AlABAMA Trial. The intervention will be centred around functionality to be incorporated into SystemOne (an electronic medical record system widely used in general practice) but the intervention package is not necessarily suitable for use in other areas of England, the UK or wider. The contextual factors to delivery should be considered. Moreover, it is only a small group of patients (around 25%–30%) who are suitable to undergo the abbreviated test (patient history, ST and OCT). Many will still require full assessment by an immunologist or allergist as per current guidelines. Cost-effectiveness analysis of the PAT and intervention package as a whole will be carried out the in upcoming AlABAMA Trial.

## CONCLUSIONS
Current clinical practice involves referral to PAT with little attention to other elements of the pathway that help to ensure that testing impacts positively on patient care. This study presents the development of a behavioural intervention package to support the process of amending incorrect penicillin allergy records. Numerous barriers to the uptake of PAT have been identified as well as penicillin prescribing and consumption following a negative test.

We have identified relevant behaviour change techniques to inform the development of the AlABAMA intervention package to overcome these barriers. The intervention is currently being tested in a feasibility trial in primary care to lead on to a randomised controlled trial.

**Collaborators** The AlABAMA team also included Jenny Boards, Mandy East, Philip Howard, Bethany Shinkins, Robert West, Ly-Mee Yu.

**Contributors** JS, SP, CCB, SS, LS, EB and ST-C designed and developed the Allergy Antibiotics and Microbial resistance programme and the plan for the intervention. MW and ST-C led the intervention development, with input from other coauthors. MW conducted the rapid review and qualitative work. MS led the behavioural analysis, logic modelling and guiding principles. CCB, EB, MD, CP and JF led the design of the trial and trial materials. MS drafted the manuscript. All authors critically reviewed the manuscript and approved the final version.

**Funding** This study summarises independent research funded by the National Institute for Health Research (NIHR) under its Programme Grants for Applied Research Programme (Grant Reference Number RP-PG-1214-20007). ST-C received additional funding from the NIHR Health Protection Research Unit (HPRU) in Healthcare Associated Infections and Antimicrobial Resistance at the University of Oxford in partnership with Public Health England [HPRU-2012-10041]. The research is supported by the NIHR infrastructure at Leeds.

**Disclaimer** The views expressed are those from the authors and not necessary those of the NHS, National Institute for Health Research or the Department of Health and Social Care.

**Competing interests** The authors have received funding from the National Institute for Health Research.

**Patient consent for publication** Not required.

**Ethics approval** This study received ethical approval from the London Bridge Research Ethics Committee (Ref: 19/LO/0176).

**Provenance and peer review** Not commissioned; externally peer reviewed.

**Data availability statement** Data sharing not applicable as no datasets generated and/or analysed for this study. No additional data available.

**ORCID iDs**
Marta Santillo http://orcid.org/0000-0001-6345-7612
Emily Bongard http://orcid.org/0000-0001-5957-6280
Sinisa Savic http://orcid.org/0000-0001-7910-0554
Sarah Tonkin-Crine http://orcid.org/0000-0003-4470-1151

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
