## [Reviewer comments · BMJ Open]

ARTICLE DETAILS

TITLE (PROVISIONAL)	Developing a behavioural intervention package to identify and amend incorrect penicillin allergy records in UK general practice and subsequently change antibiotic use
AUTHORS	Santillo, Marta; Wanat, Marta; Davoudianfar, Mina; Bongard, Emily; Savic, Sinisa; Savic, Louise; Porter, Catherine; Fielding, Joanne; Butler, Christopher C.; Pavitt, Sue; Sandoe, Jonathan; Tonkin-Crine, Sarah

VERSION 1 - REVIEW

REVIEWER	Dr Caroline Coope University of Bristol, UK
REVIEW RETURNED	13-Apr-2020

GENERAL COMMENTS	The article describes a mixed method three-stage research study that iteratively informed the development of an intervention aimed at supporting primary care clinicians and patients to refer and attend for, respectively, penicillin allergy testing (PAT), and if negative to support the prescription and consumption of penicillin. The authors claimed to use an integrative approach to intervention development that incorporated theory, evidence and a person-based approach. Stage 1 included a rapid review, qualitative studies and expert consultations. Stage 2 aimed at developing the intervention and included developing guiding principles for the intervention development, a behavioural analysis and logic framework to map the key components of the intervention onto the 'needs' of patients and clinicians and the change processes of the intervention. Stage 3 utilised think-aloud sessions with clinicians and patients to optimise the intervention materials. Reviewer comments on review checklist 'No' answers 2. Is the abstract accurate, balanced and complete? The 'Results' section of the abstract does not cover results for each of the three research objectives. Currently results related to barriers identified regarding objectives 1 & 2 are presented. Results related to barriers to the prescription and consumption of first-line penicillin following a negative test result (objective 3) should also be presented here for completeness of the abstract. 4. Are the methods described sufficiently to allow the study to be repeated? This paper incorporates a synthesis of the results of two other studies (a rapid review and qualitative study) where the methods are reported elsewhere in detail. The methods for these two studies are not described sufficiently here however it is customary practice to
---

reference, as here, to detail of methods elsewhere if published in peer review articles. However, the authors do state that two papers were reviewed for the rapid review containing patients views and so should balance this by stating how many papers reviewed reflected clinicians' views.

The acronym used for the intervention is not consistent. [page 5, line 23] "ALABAMA" intervention package. [page 5, line 31 and line 40] "AIBAMA" programme. Are they different or is this a typo? Please correct whichever is incorrect and then throughout the document. A description of the methods relating to the primary research presented in this paper is not complete to the degree that this work could be repeated. See details below:

[page 6, line 47] "An inductive thematic analysis approach" ...is a method and not a result. Please put this in the methods section and reference this approach.

[page 8, line 15-19] as follows: "Guiding principles were then created to outline the intervention objectives and the key design features which addressed them." How were these created?

[page 11, line 8-12] Is the resource you designed the same as the 'pre-intervention booklet' in S1 intervention component? If yes, it would help the reader if you use the same terminology in the text as in the table to link them more obviously.

[page 11, lines 49-51] Authors assume a high level of knowledge of the reader about the TDF and BCW. A clear explanation of how the TDFs, BCW and BCTs fit together may help here. For instance, when mentioning relevant BCTs social support and restructuring of the social and physical environment, what theoretical constructs do each of these pertain too?

[page 11, line 51] "as part of the trial". What trial? Did you pilot the materials or is there a trial alongside this study?

For stage 3 optimising the intervention materials, a clearer description of how you undertook the Think Aloud interviews is required. Did you use a topic guide or follow the materials structure or both? Please describe details of the methods used.

5. Are research ethics (e.g. participant consent, ethics approval) addressed appropriately?

For the primary research Think Aloud interviews, there is no mention of any information about the study that was given to the participants or how you gained consent to conduct the research.

6. Are the outcomes clearly defined?

When describing the logic model you refer to the behavioural outcomes (p13, line 35-36) and then the trial outcome (line 37-38). I don't think you need to present details of a trial outcome here. You previously state that the trial will happen after this work and you do not state that an objective of this intervention development is to develop a trial outcome, so it seems misplaced here.

10. Are they (results) presented clearly?

It is common practice in academic articles for there to be a methods section and then a results section. Can you justify why you have presented you're Results following each description of methods? In addition, you have labelled the 'Results' as 'Outcomes' throughout this paper e.g. p6, line 3 and p6 line 45. This is confusing. Please change the sub-titles to reflect the content of the narrative presented i.e. key results.

There are no results presented for the expert discussions? The findings from the expert discussion informed the intervention and therefore it is important to include these as results.

11. Are the discussion and conclusions justified by the results

The discussion requires further work to improve its cohesiveness as it currently includes the use of different tenses as if cut and pasted

	together from separate pieces of work. In addition, some of the sentences are not complete. First sentence of the discussion (p18, line54) is not clear i.e. “We the development”? This is also a very long sentence and would benefit from being divided into two shorter sentences. P19, line24 ‘in patient’ should be one word ‘inpatient’. The discussion talks about the development of trial outcomes, however this was not an objective stated in this intervention study. I would therefore remove this. P20, line12-22. This states that the intervention explained which psychological mechanisms were changed by the intervention, and which target behaviours affected the intervention primary outcome (the treatment “response failure”). You have not tested the effect of the intervention and so these are theorised changes rather than tested. The primary outcome relates to the future trial and so why are you reporting here as if it has been conducted? Line 41 confirms it has been? P20 Line 54-54 States the trial is upcoming. So which is it? Completed or upcoming? P20, line 43: What is System One? Please explain.
--	--

REVIEWER	Liz Steed QMUL, UK
REVIEW RETURNED	03-Jun-2020

GENERAL COMMENTS	This study describes the development of a complex intervention (directed at both clinician and patient behaviours) to identify and overcome incorrect labeling of patients as penicillin allergic. I find this a very well written paper that describes in detail the process of intervention development for the ALABAMA intervention. Inevitably these papers are complex and include a lot of detail. I would only recommend a couple of required changes before publication, but have also included some suggestions for the authors to consider which I feel may help the reader get the most from the paper. Required changes Reporting standards for intervention development have now been published (GUIDED, 2020). I would expect this paper to report their adherence to these guidelines, at minimum as a supplementary document. Within the TiDieR description and at certain points within the document there appears to be description of research processes (E.g. recruitment) under the description of intervention components. These are distinct processes and should be recognized as such. If the authors feel it necessary to include research processes in the TiDieR intervention description perhaps they could at minimum highlight this through the use of italics or some other indicator. Suggestions for authors to consider. My main comment relates to the description of the behaviours to be changed, and the use of terminology for these, which seems to vary through the document and hence makes an already complex topic even more complex. For example the title reports the intervention aims to ‘identify and amend incorrect penicillin allergy records.’ This could be interpreted as something as simple as looking at medical
---

records and changing the e-records on the computer. However ALABAMA appears to be doing something far greater with four key behaviours , two for patients (P) and two for clinicians (C)

- P1) attendance at Penicillin Allergy Testing (PAT);
- P2) Patient consumption of penicillin when prescribed first line after a negative test,
- C1) Clinician referral of low risks patients to Penicillin Allergy Testing;
- C2) Clinician prescription of penicillin after negative test result

I wonder if a title which talks about 'identify and overcoming/responding (or something similar) to incorrect penicillin allergy records' might be more reflective of what is being done

It may also aid comprehension if the behaviours are consistently presented throughout as either clinician first then patient or vice versa. At the moment it tends to vary within tables and text which can add to confusion.

Logic Model. Similar to the comment above you currently talk about referral of low risk patients and their attendance as 'assessment of incorrect penicillin records' does this equate to 'identify' (ie the referral and uptake of PAT). Using the same term again may simplify things. Similarly adding the term 'overcoming/responding' as an intervention target would seem to follow your logic, whilst 'The PAT is not efficient and cost-effective' does not really seem to have been addressed or identified in the text, so feels like a bit of an anomaly. Could the later be addressed by adding this dimension to the text or if not such an important aspect of the final intervention removing?

If the above changes were made the logic model may need rejigging so at each level of intervention target (identify and overcome/respond) there are both clinician and patient behaviours) identify – P1,C1; overcome P2,C2. This could potentially follow through to purported mediators, where again 'change of incorrect labels' doesn't seem representative enough of what is truly happening which is something more like response to changed labeling

Do the intervention processes labeled in the logic model essentially link to the theoretical domains from the TDF. I think it would be helpful to indicate this if so.

Minor thoughts

It might be helpful to note in the background that penicillin allergy can relate to a whole set of penicillin related antibiotics.

Patient and public involvement - Generally participants enrolled in a research study (regardless of whether it is qualitative or quantitative) are not considered PPI. It would be more accurate to report those elements that are truly PPI and perhaps if desired then state that the research methodology (interviews etc) further specifically aimed to understand the patient perspective

Rapid review – full methods and results don't need to be gone in to but it would be useful to know at least the methodology of the two studies for patients. Also how many papers were there on clinician's views (were these in fact the same papers as the patients)

	Qualitative study - was the TDF or any of the other behavioural frameworks used to inform the semi-structured interview Expert study – did this expert group include the health psychologist member of your team. It is important to understand if this level of expertise was being received at this point given the critical phases of developing guiding principles and applying theory I am unclear which way around the mapping of theory to intervention components occurred. Were the components devised and then mapped to ensure they had the right content (this is how it appears to be presented) or was the theory mapped to barriers for each behavior which then informed the intervention components. This would be helpful to be clarified and might have implications for how table S1 is laid out. In the think aloud methods was a formal application of this methodology used (in which case it may be useful to describe this a bit more) or was it more an over arching interview receiving feedback on intervention components The changes to patient information letters is a small paragraph at the end of the paper but actually this is quite important from an implementation perspective – this could be picked up on in the discussion? Overall the discussion appears the weaker section of the paper, it may be helpful to particularly revisit the opening paragraph. Figures These are quite difficult to see but would the behavioural analysis not have informed the intervention component independent of through the logic model? Also is it the logic model that informs the think aloud interviews. It is appreciated that trying to represent an iterative process through linear diagrams is difficult but perhaps this could at least be reflected upon within the discussion.
--	---

VERSION 1 – AUTHOR RESPONSE

Reviewer: 1

Comment 1:

The article describes a mixed method three-stage research study that iteratively informed the development of an intervention aimed at supporting primary care clinicians and patients to refer and attend for, respectively, penicillin allergy testing (PAT), and if negative to support the prescription and consumption of penicillin. The authors claimed to use an integrative approach to intervention development that incorporated theory, evidence and a person-based approach. Stage 1 included a rapid review, qualitative studies and expert consultations. Stage 2 aimed at developing the intervention and included developing guiding principles for the intervention development, a behavioural analysis and logic framework to map the key components of the intervention onto the 'needs' of patients and clinicians and the change processes of the intervention. Stage 3 utilised think-aloud sessions with clinicians and patients to optimise the intervention materials.

Response: We thank the reviewer for the detailed comments and suggestions that have improved the manuscript.

Comment 2:

The 'Results' section of the abstract does not cover results for each of the three research objectives. Currently results related to barriers identified regarding objectives 1 & 2 are presented. Results related to barriers to the prescription and consumption of first-line penicillin following a negative test result (objective 3) should also be presented here for completeness of the abstract.

Response: Thank you for this comment. We have added the barriers identified for prescription and consumption of first-line penicillin following a negative test result (objective 3) to the abstract.

Lines 40-42: "The key barriers to the prescription and consumption of first-line penicillin following a negative test result were patient and clinician beliefs about the accuracy of PAT and whether taking penicillin was safe."

Comment 3:

This paper incorporates a synthesis of the results of two other studies (a rapid review and qualitative study) where the methods are reported elsewhere in detail. The methods for these two studies are not described sufficiently here however it is customary practice to reference, as here, to detail of methods elsewhere if published in peer review articles. However, the authors do state that two papers were reviewed for the rapid review containing patients views and so should balance this by stating how many papers reviewed reflected clinicians' views.

Response: Thank you for the comment. We have added the number of papers identified which reflected clinicians' views.

Lines 123-124: The review also identified six studies on clinicians' views of PAT, three of which specifically focused on barriers and facilitators to using PAT.

Comment 4:

The acronym used for the intervention is not consistent. [page 5, line 23] "ALABAMA" intervention package. [page 5, line 31 and line 40] "AIABAMA" programme. Are they different or is this a typo? Please correct whichever is incorrect and then throughout the document.

Response: Thank you for identifying this typo. We have now correct them through the document with the right acronym "AIABAMA".

Comment 5:

[page 6, line 47] "An inductive thematic analysis approach"...is a method and not a result. Please put this in the methods section and reference this approach.

Response: Thank you for the comment. We have moved the sentence to the methods section and added a reference.

Lines 139-140: "An inductive thematic analysis approach was used to analyse data."

Comment 6:

[page 8, line 15-19] as follows: "Guiding principles were then created to outline the intervention objectives and the key design features which addressed them." How were these created?

Response: Thank you for your comment. We have provided more details regarding how the guiding principles were created.

Lines 179-182: "As a first step to creating guiding principles, characteristics and behavioural needs of the target users were identified based on the findings of the rapid review, qualitative interviews and expert discussions. The second step involved identification of intervention objectives and key design features which would address these needs."

Comment 7:

[page 11, line 8-12] Is the resource you designed the same as the 'pre-intervention booklet' in S1 intervention component? If yes, it would help the reader if you use the same terminology in the text as in the table to link them more obviously.

Response: Thank you for your comment. We have amended the text and used the same terminology throughout. We have used "Penicillin Allergy Testing: Information for general practice" leaflet for the clinician resource. We have used "Penicillin allergy testing: going for a test" booklet and "Penicillin allergy testing: a negative test result" booklet for the two resources for patients.

Comment 8:

[page 11, lines 49-51] Authors assume a high level of knowledge of the reader about the TDF and BCW. A clear explanation of how the TDFs, BCW and BCTs fit together may help here. For instance, when mentioning relevant BCTs social support and restructuring of the social and physical environment, what theoretical constructs do each of these pertain too?

Response: Thank you for your comment. We have provided an explanation of how TDF, BCW and BCTTv1 fit together.

Lines 199-205: "We used a model of behaviour known as COM-B (Capability, Opportunity, Motivation, Behaviour)²². According to this model, behaviour is a result of these three constructs.. Changing behaviour will require a change in these three constructs. The Behaviour Change Wheel (BCW)²³ offers a systematic way to identify types of intervention component which are best suited to changing each of these components. A more specific description of elements which can be delivered within intervention components is provided in a taxonomy of 93 behaviour change techniques (Behaviour Change Technique Taxonomy, v1; BCTT v1)²⁴."

Lines: 208-218: "The first step of the behavioural analysis process was to define the problem in behavioural terms which resulted in the identification of the four specific target behaviours. As a second step, we identified what needed to change using the Theoretical Domain Framework(TDF)²⁵ which includes 14 domains such as knowledge, skills, beliefs about capabilities. Each domain of the TDF relates to a COM-B component. This step allowed us to identify target behaviours, their barriers and facilitators, and how intervention components would support desired behaviour change based on evidence collated in stage 1. Finally, the intervention components were mapped to the TDF framework and the BCW) referring to the BCTv1²⁴. This produced a list of TDF barriers to the target behaviours, target constructs (what needs to change for the behaviour to occur), intervention functions (ways an intervention can change behaviour) and behaviour change techniques used for each of the barriers/facilitators."

Comment 9:

[page 11, line 51] "as part of the trial". What trial? Did you pilot the materials or is there a trial alongside this study?

Response: Thank you for the comment. We have added a sentence in the background to provide more information about the study trial.

Lines 87-90: “The AIABAMA intervention will be trialled in a national randomised controlled trial in winter 2020-2021. It will test whether changes removal of incorrect penicillin allergy records and subsequent penicillin prescribing will affect treatment response for patients taking antibiotics for infections.”

Comment 10:

For stage 3 optimising the intervention materials, a clearer description of how you undertook the Think Aloud interviews is required. Did you use a topic guide or follow the materials structure or both? Please describe details of the methods used.

Response: Thank you for your comment. We have added a description of how we undertook the Think Aloud interviews and we provide a copy of the topic guide.

Lines 309-311: “Interviews were not recorded, but the interviewer took detailed notes. Interviews focused on GP views of each section of the “Penicillin Allergy Testing: Information for general practice” (supplementary materials S2).”

Lines 337-340: “Again, interviews were not recorded, but the interviewer took detailed notes. Interviews asked their views about the two patient booklets (“Penicillin Allergy Testing: going for a test”, “Penicillin Allergy testing: a negative test result”) and the intervention card(supp doc X).”

Comment 11:

Are research ethics (e.g. participant consent, ethics approval) addressed appropriately? For the primary research Think Aloud interviews, there is no mention of any information about the study that was given to the participants or how you gained consent to conduct the research.

Response: Thank you for your comment. Think-aloud interviews were only carried out to refine intervention materials and as such were not classed a distinct piece of research requiring ethical approval.

Lines 297-300: “Think-aloud interviews were carried out as part of intervention development and not as a piece of qualitative research, as such ethical approval was not required. Interviews were not recorded and interviews were only used to refine intervention materials.”

Comment 12:

When describing the logic model you refer to the behavioural outcomes (p13, line 35-36) and then the trial outcome (line 37-38). I don't think you need to present details of a trial outcome here. You previously state that the trial will happen after this work and you do not state that an objective of this intervention development is to develop a trial outcome, so it seems misplaced here.

Response: Thank you for your comment. In the logic model we have deleted the trial outcome and only report the behavioural outcome (“the prescription and consumption of first-line penicillin when indicated”).

Comment 13:

It is common practice in academic articles for there to be a methods section and then a results section. Can you justify why you have presented you're Results following each description of methods? In addition, you have labelled the 'Results' as 'Outcomes' throughout this paper e.g. p6, line 3 and p6 line 45. This is confusing. Please change the sub-titles to reflect the content of the narrative presented i.e. key results.

Response: Thank you for your comment. As you and your colleagues have recognised, this is a complex paper which describes the three stages of intervention development. Each of these three stages include multiple steps (for stage 1: rapid review, qualitative interviews, expert discussions, for stage 2: guiding principles, behavioural analysis, logic model; stage 3: think aloud interviews). We decided to present the methods and results of each step/ stage to make the manuscript easier to follow. In addition to this, intervention development is an iterative process and the outcome of one of these steps can inform the methods of the following step. We have now relabelled the headings previously called 'Outcomes' to 'Results' throughout the manuscript.

Comment 14:

There are no results presented for the expert discussions? The findings from the expert discussion informed the intervention and therefore it is important to include these as results.

Response: Thank you for the comment. We have now presented the results for the expert discussions.

Lines 170-172: "The expert discussions informed the interpretation of the evidence collated in the rapid review and the qualitative studies, the development of early iterations of the intervention materials, and the development of the initial intervention plan and components."

Comment 15:

The discussion requires further work to improve its cohesiveness as it currently includes the use of different tenses as if cut and pasted together from separate pieces of work. In addition, some of the sentences are not complete.

Response: Thank you for your comment. We have worked on the Discussion to improve its cohesiveness and checked the grammar throughout.

Comment 16:

First sentence of the discussion (p18, line54) is not clear i.e. "We the development"? This is also a very long sentence and would benefit from being divided into two shorter sentences.

Response: Thank you. We have changed the typo in the first sentence of the Discussion and have divided the sentence in to two.

Lines 388-391: "This paper describes the development of the AIABAMA intervention package, which aimed to change behaviours that facilitate penicillin allergy testing and subsequently amend incorrect allergy records in English general practice. Such changes in clinician and patient behaviours have potential to significantly impact on antibiotic prescribing and consumption."

Comment 17:

P19, line24 'in patient' should be one word 'inpatient'.

Response: Thank you for identifying the typo. We have change 'in patient' to 'inpatient'.

Comment 18:

The discussion talks about the development of trial outcomes, however this was not an objective stated in this intervention study. I would therefore remove this.

Response: Thank you for the suggestion. We have removed the development of trial outcomes from the discussion.

Comment 19:

P20, line12-22. This states that the intervention explained which psychological mechanisms were changed by the intervention, and which target behaviours affected the intervention primary outcome (the treatment “response failure”). You have not tested the effect of the intervention and so these are theorised changes rather than tested.

Response: Thank you for your comment. We have changed how we report the psychological mechanisms of the intervention and refer to them as hypothesised effects on the outcomes.

Lines 404-410: “More specifically it highlighted the BCTs used in each component of the AIABAMA intervention package and how they addressed the barriers identified in the rapid review and qualitative work. It also explained which psychological mechanisms were hypothesised to change by the intervention components (clinicians’ knowledge on penicillin allergy and PAT procedures, patients’ beliefs about positive consequences of taking the PAT) and which target behaviours (referral and attendance to PAT, prescription and consumption of penicillin first line) would likely affect the intervention outcomes.”

Comment 20:

The primary outcome relates to the future trial and so why are you reporting here as if it has been conducted? Line 41 confirms it has been? P20 Line 54-54 States the trial is upcoming. So which is it? Completed or upcoming?

Response: Thank you for your comment. We have changed the tense in the Discussion when referring to the trial, which will be conducted in 2020-2021.

Lines 430-435: “The AIABAMA programme will focus on one geographical area (the North of England), which will be covered by the specialist allergy unit participating in the AIABAMA trial. The intervention will be centred around functionality to be incorporated into SystemOne (an electronic medical record system widely used in general practice) but the intervention package is not necessarily suitable for use in other areas of England, the UK or wider.”

Comment 21:

P20, line 43: What is System One? Please explain.

Response: Thank you for your comment. We have provided a brief explanation of what SystemOne is.

Lines 432-434: “The intervention will be centred around functionality to be incorporated into SystemOne (an electronic medical record system widely used in general practice)”.

Reviewer: 2

This study describes the development of a complex intervention (directed at both clinician and patient behaviours) to identify and overcome incorrect labelling of patients as penicillin allergic. I find this a very well written paper that describes in detail the process of intervention development for the AIABAMA intervention. Inevitably these papers are complex and include a lot of detail. I would only

recommend a couple of required changes before publication, but have also included some suggestions for the authors to consider which I feel may help the reader get the most from the paper.

Response: Thank you so much for the positive comment. We have amended the manuscript in regards to your required and suggested changes which has improved the clarity of the manuscript.

Required changes

Comment 22:

Reporting standards for intervention development have now been published (GUIDED, 2020). I would expect this paper to report their adherence to these guidelines, at minimum as a supplementary document.

Response: Thank you for your comment. We have now completed the GUIDED checklist (supplementary material S3). We have referred to it in the main text.

Lines 380-381: "We have followed the GUIDED checklist (supplementary materials S4), when reporting intervention development in this paper."

Comment 23:

Within the TiDieR description and at certain points within the document there appears to be description of research processes (E.g. recruitment) under the description of intervention components. These are distinct processes and should be recognized as such. If the authors feel it necessary to include research processes in the TiDieR intervention description perhaps they could at minimum highlight this through the use of italics or some other indicator.

Response: Thank you for your suggestion. We have removed the description of trial research processes as part of description of the intervention components in the TiDieR description.

Suggestions for authors to consider.

Comment 24:

My main comment relates to the description of the behaviours to be changed, and the use of terminology for these, which seems to vary through the document and hence makes an already complex topic even more complex. For example the title reports the intervention aims to 'identify and amend incorrect penicillin allergy records.' This could be interpreted as something as simple as looking at medical records and changing the e-records on the computer. However AIABAMA appears to be doing something far greater with four key behaviours, two for patients (P) and two for clinicians (C)

- P1) attendance at Penicillin Allergy Testing (PAT);
- P2) Patient consumption of penicillin when prescribed first line after a negative test,
- C1) Clinician referral of low risks patients to Penicillin Allergy Testing;
- C2) Clinician prescription of penicillin after negative test result

I wonder if a title which talks about 'identify and overcoming/responding (or something similar) to incorrect penicillin allergy records' might be more reflective of what is being done.

Response: Thank you for this comment. We have described the four target behaviours throughout the text of the manuscript and tables as:

- 1) Referral of patients for penicillin allergy testing (PAT),

- 2) attendance for PAT and
- 3) prescription or consumption penicillin, when indicated, following a negative PAT result.

We have changed the title to be more reflective of the four behaviours: Developing a behavioural intervention package to identify and amend incorrect penicillin allergy records in UK general practice and subsequently change antibiotic use.

Comment 25:

It may also aid comprehension if the behaviours are consistently presented throughout as either clinician first then patient or vice versa. At the moment it tends to vary within tables and text which can add to confusion.

Response: Thank you for this suggestion. We have changed the presentation of the behaviours. We have presented the clinician behaviours first and then patient's behaviours through the text and tables.

Comment 26:

Logic Model. Similar to the comment above you currently talk about referral of low risk patients and their attendance as 'assessment of incorrect penicillin records' does this equate to 'identify' (ie the referral and uptake of PAT). Using the same term again may simplify things. Similarly adding the term 'overcoming/responding' as an intervention target would seem to follow your logic, whilst 'The PAT is not efficient and cost-effective' does not really seem to have been addressed or identified in the text, so feels like a bit of an anomaly. Could the later be addressed by adding this dimension to the text or if not such an important aspect of the final intervention removing?

Response: Thank you for the suggestion. We have removed 'assessment of incorrect penicillin labels' and 'PAT is not efficient and cost-effective' from the problem and outcome columns as they refer to the trial, rather than the behavioural package.

Comment 27:

If the above changes were made the logic model may need rejigging so at each level of intervention target (identify and overcome/respond) there are both clinician and patient behaviours) identify – P1,C1; overcome P2,C2. This could potentially follow through to purported mediators, where again 'change of incorrect labels' doesn't seem representative enough of what is truly happening which is something more like response to changed labelling

Response: Thank you for your comment. We have changed the logic model so that the intervention targets include all four behaviours and we have modified the purported mediators column removing 'changing of incorrect labels'.

Comment 28:

Do the intervention processes labelled in the logic model essentially link to the theoretical domains from the TDF. I think it would be helpful to indicate this if so.

Response: Thank you for the comment. We have explained in the main text of the manuscript that the processes in the logic model link to the TDF.

Lines 262-265: “These were the psychological factors which explained the relationships between the intervention components and the outcome of the intervention. These processes were described in terms of TDF domains. Each intervention technique was hypothesised to mainly affect one of these processes.”

Minor thoughts

Comment 29:

It might be helpful to note in the background that penicillin allergy can relate to a whole set of penicillin related antibiotics.

Response: Thank you for the suggestion. We have added a line to the Background explaining that penicillin allergy relates to a whole group of antibiotics (penicillins).

Lines 67-69: “Penicillin allergy refers to an allergy for a whole group of antibiotics (penicillins) which are first line antibiotics for many common infections. As a result of incorrect penicillin allergy labels, a significant portion of the population is denied effective antibiotics.”

Comment 30:

Patient and public involvement - Generally participants enrolled in a research study (regardless of whether it is qualitative or quantitative) are not considered PPI. It would be more accurate to report those elements that are truly PPI and perhaps if desired then state that the research methodology (interviews etc) further specifically aimed to understand the patient perspective.

Response: Thank you for the comment. We have now reported only strictly PPI activities in the Public and Patient Involvement section. In the method section of the think-aloud paragraph we have explained that some PPI members took part in the interviews.

Lines 101-104. “The AIABAMA programme grant involved PPI members, as co-applicants, and study advisors from the start so research questions were informed by their input. Results of all work from the AIABAMA programme will be disseminated with the help from our PPI co-apps and our existing PPI networks towards the end of the programme grant.”

Lines 335-337: “These participants were asked at the end of the initial interview whether they would like to comment on the intervention materials at a later date; the additional two participants were identified using convenience sampling.”

Comment 31:

Rapid review – full methods and results don't need to be gone in to but it would be useful to know at least the methodology of the two studies for patients. Also how many papers were there on clinician's views (were these in fact the same papers as the patients).

Response: Thank you for this comment, which is similar to Comment 3 from reviewer 1. We have added how many papers reflected clinicians' views.

Lines 123-125: “The review also identified six studies on clinicians' views of PAT, three of which specifically focused on barriers and facilitators to using PAT.”

Comment 32:

Qualitative study - was the TDF or any of the other behavioural frameworks used to inform the semi-structured interview

Response: No, the topic guides used in the qualitative research were not informed by the TDF specifically but instead by the existing literature which indicated potential barriers and facilitators to target behaviours. We have added this to the main text.

Line 138-139: "Interview topic guides were informed by existing literature which indicated potential barriers and facilitators to target behaviours."

Comment 33:

Expert study – did this expert group include the health psychologist member of your team. It is important to understand if this level of expertise was being received at this point given the critical phases of developing guiding principles and applying theory

Response: Thank you for your comment. The Expert discussions took place with the wider clinical team which included "a consultant immunologist, a consultant microbiologist, a consultant anaesthetist, a general practitioner and professors with expertise in applied health research". The Health Psychologist was part of the core research team for this task, the research team also contributed to these discussions.

Comment 34:

I am unclear which way around the mapping of theory to intervention components occurred. Were the components devised and then mapped to ensure they had the right content (this is how it appears to be presented) or was the theory mapped to barriers for each behaviour which then informed the intervention components. This would be helpful to be clarified and might have implications for how table S1 is laid out.

Response: Thank you for your comment. The intervention development was an iterative approach which was informed by evidence, theory and consultation/interviews with target group representatives. We have provided more explanation on the iterative approach in the main text. The order of the columns in Table S1 do not represent a linear approach but they are the standard layout used by the creators of the Person-Based Approach. The first three columns of the table are specific to the intervention and the following columns show the mapping of each barrier/intervention ingredient to the theory.

Comment 35:

In the think aloud methods was a formal application of this methodology used (in which case it may be useful to describe this a bit more) or was it more an overarching interview receiving feedback on intervention components

Response: Thank you for this comment. We have provided more information on the methods of the think aloud interviews.

Lines: 306-311. “Four of the participants had previously taken part in the qualitative study on GP views on the barriers and facilitators for PAT and antibiotic use after a negative test (mentioned in the section Qualitative work above Ref]; four additional participants were recruited using convenience sampling. The interviews were not recorded, but the interviewer took detailed notes. These interviews focused on GP views of each section of the “Penicillin Allergy Testing: Information for general practice” and did not cover their general views on penicillin allergy or penicillin allergy testing.”

Lines: 333-340: “Five of these patients had previously taken part in the qualitative study on views on the barriers and facilitators for PAT and antibiotic use after a negative test (mentioned in the section Qualitative work above Ref]. These participants were asked at the end of the initial interview whether they would like to comment on the intervention materials when they are ready; the additional two participants were identified using convenience sampling. The interviews were not recorded, but the interviewer took detailed notes. Interviews asked their views about the two patient booklets (“Penicillin Allergy Testing: going for a test”, “Penicillin Allergy testing: a negative test result”) and the intervention card and did not include any questions about their personal views or experiences of penicillin allergy.”

Comment 36:

The changes to patient information letters is a small paragraph at the end of the paper but actually this is quite important from an implementation perspective – this could be picked up on in the discussion?

Response: Thank you for this comment. We have expanded the paragraph on the optimisation of intervention materials and moved it as first paragraph of stage 3.

Lines 290-300: “The development of the intervention materials was part of an iterative process which was initially informed by the findings of the rapid review, qualitative interviews and expert consultations. Once the first drafts of the intervention materials were developed, a series of think-aloud interviews with clinicians and patients were conducted in order to gain their insights on the content, usability and format of the clinician “Penicillin Allergy Testing: Information for general practice” leaflet and two patient booklets (“Penicillin Allergy Testing: going for a test”, “Penicillin Allergy testing: a negative test result”). Subsequently to the feedback received by the participants in the think-aloud interviews and discussions in further expert consultations, the intervention materials were optimised.”

Comment 37:

Overall the discussion appears the weaker section of the paper, it may be helpful to particularly revisit the opening paragraph.

Response: Thank you for the comment. We have revised the opening paragraph of the Discussion.

Lines 388-391: “This paper describes the development of the AIABAMA intervention package, which aimed to change behaviours that facilitate penicillin allergy testing and subsequently amend incorrect allergy records in English general practice. Such changes in clinician and patient behaviours have potential to significantly impact on antibiotic prescribing and consumption.”

Comment 38:

Figures

These are quite difficult to see but would the behavioural analysis not have informed the intervention component independent of through the logic model? Also is it the logic model that informs the think aloud interviews. It is appreciated that trying to represent an iterative process through linear diagrams is difficult but perhaps this could at least be reflected upon within the discussion.

Response: Thank you for your comment. We have now changed the figures to reflect the iterative process in more details for those examples.

VERSION 2 – REVIEW

REVIEWER	Dr Liz Steed Queen Mary University of London
REVIEW RETURNED	18-Jul-2020

GENERAL COMMENTS	The authors have taken on board the comments made by reviewers, however there are a few sentences/paragraphs where new text has been added which need reviewing. Assuming the authors do this I will be happy for publication Check wording of added sentence pg 5 133-6, it currently is unclear Pg 7 the last para has rather a lot of acronyms making it difficult to understand Under expert discussion. I appreciate your emphasis is on the wider clinical team – do you actually say who is in your core team anywhere. You could address this here by saying something like ‘in addition to our core team (which included xx & x) the wider team.... On page 10 – you say changing behavior will require a change in these three constructs (capability, opportunity, motivation) this is not strictly true for example if someone has all the capability and motivation in the world but not opportunity then the behavior will break down so change only needs opportunity to change. You just need to slightly rephrase for example something like ‘changing behavior requires overcoming any presenting barriers in each of these areas’. In fact this whole paragraph would benefit from review to ensure it’s accuracy (again the BCW – is a systematic way to develop interventions). Think-aloud interviews were carried out as part of intervention development and not as a piece of qualitative research, as such ethical approval was not required – I am not happy with the way this sentence now reads as it could lead others to think that this sort of approach does not need ethical approval. I think the point you are trying to make is that ‘a think aloud approach was used with members of the PPI group(?) for the aim of feedback on intervention materials, as this did not constitute a
---

	formal qualitative evaluation ethical approval was not required or sort. You might want to consider whether the word interview is helpful as this does imply a more formal study than something like feedback
--	---

VERSION 2 – AUTHOR RESPONSE

Reviewer 2

Comment 1. Check wording of added sentence pg 5 133-6, it currently is unclear

Response: We thank the reviewer for the comment. We have corrected the sentence.

Lines 87-88: "It will test how correction of incorrect penicillin allergy records and subsequent penicillin prescribing influence recovery in patient taking antibiotics for infections."

Comment 2: Pg 7 the last para has rather a lot of acronyms making it difficult to understand

Response: We thank the reviewer. We have now changed the sentence and used fewer acronyms to make the sentence easier to understand.

Lines 154-160: Moreover, when the test was perceived as more invasive, e.g. the oral challenge test compared to the skin test, patients reported being more concerned about PAT. Patients were also concerned about how much they would be monitored during the test and highlighted the importance of being informed, kept safe and observed by qualified professionals.

Comment 3: Under expert discussion. I appreciate your emphasis is on the wider clinical team – do you actually say who is in your core team anywhere. You could address this here by saying something like 'in addition to our core team (which included xx & x) the wider team....

Response: We thank the reviewer. We have added a sentence to explain who is part of the core team.

Lines 165-169: In addition to our core team (which included three psychologists), the wider clinical research team included a consultant immunologist, a consultant microbiologist, a consultant anaesthetist, a general practitioner and professors with expertise in applied health research. As part of this first stage we consulted our wider clinical research team using monthly tele-conferences and emails to gain their feedback on several aspects of the intervention development.

Comment 4: On page 10 – you say changing behavior will require a change in these three constructs (capability, opportunity, motivation) this is not strictly true for example if someone has all the capability and motivation in the world but not opportunity then the behavior will break down so change only needs opportunity to change. You just need to slightly rephrase for example something like 'changing behavior requires overcoming any presenting barriers in each of these areas'.

Response: We thank you the reviewer for the comment. We have reviewed the sentence and changed it as suggested.

Lines 206-207: According to this model, behaviour is a result of these three constructs. Changing behaviour requires overcoming any presenting barriers in each of these areas.

Comment 5: In fact this whole paragraph would benefit from review to ensure it's accuracy (again the BCW – is a systematic way to develop interventions).

Response: We thank the reviewer, we have corrected the paragraph as suggested.

Lines 207-209: The Behaviour Change Wheel (BCW)²³ offers a systematic way to develop interventions to change each of these constructs.

Comment 6: Think-aloud interviews were carried out as part of intervention development and not as a piece of qualitative research, as such ethical approval was not required –
I am not happy with the way this sentence now reads as it could lead others to think that this sort of approach does not need ethical approval. I think the point you are trying to make is that 'a think aloud approach was used with members of the PPI group(?) for the aim of feedback on intervention materials, as this did not constitute a formal qualitative evaluation ethical approval was not required or sort. You might want to consider whether the word interview is helpful as this does imply a more formal study than something like feedback

Response: We thank the reviewer for this comment. We have changed the paragraph according to the suggestions. We have presented the think-aloud approach used to gain feedback on intervention materials, and we have used through the manuscript the word "think-aloud consultations" rather than "think-aloud interviews".

Lines 303-306: Once the first drafts of the intervention materials were developed, a think aloud approach was used with clinicians and members of the PPI group for the aim of feedback on intervention materials. As this did not constitute a formal qualitative evaluation ethical approval was not required.

VERSION 3 – REVIEW

REVIEWER	Liz Steed Queen Mary University of London UK
REVIEW RETURNED	10-Aug-2020

GENERAL COMMENTS	I am happy with all amendments, a very nice paper - I look forward to seeing it published
---